# Deep Learning Applied to Defect Detection in Powder Spreading Process of Magnetic Material Additive Manufacturing

**DOI:** 10.3390/ma15165662

**Published:** 2022-08-17

**Authors:** Hsin-Yu Chen, Ching-Chih Lin, Ming-Huwi Horng, Lien-Kai Chang, Jian-Han Hsu, Tsung-Wei Chang, Jhih-Chen Hung, Rong-Mao Lee, Mi-Ching Tsai

**Affiliations:** 1Department of Mechanical Engineering, National Cheng Kung University, Tainan 701, Taiwan; 2Department of Computer Science and Information, National Pingtung University, Pingtung 900, Taiwan; 3Department of Intelligent Robotics, National Pingtung University, Pingtung 900, Taiwan

**Keywords:** convolution neural network, metal additive manufacturing, powder-spreading defect, selective laser melting

## Abstract

Due to its advantages of high customization and rapid production, metal laser melting manufacturing (MAM) has been widely applied in the medical industry, manufacturing, aerospace and boutique industries in recent years. However, defects during the selective laser melting (SLM) manufacturing process can result from thermal stress or hardware failure during the selective laser melting (SLM) manufacturing process. To improve the product’s quality, the use of defect detection during manufacturing is necessary. This study uses the process images recorded by powder bed fusion equipment to develop a detection method, which is based on the convolutional neural network. This uses three powder-spreading defect types: powder uneven, powder uncovered and recoater scratches. This study uses a two-stage convolutional neural network (CNN) model to finish the detection and segmentation of defects. The first stage uses the EfficientNet B7 to classify the images with/without defects, and then to locate the defects by evaluating three different instance segmentation networks in second stage. Experimental results show that the accuracy and Dice measurement of Mask-R-CNN network with ResNet 152 backbone can reach 0.9272 and 0.9438. The computational time of an image only takes approximately 0.2197 sec. The used CNN model meets the requirements of the early detected defects, regarding the SLM manufacturing process.

## 1. Introduction

Metal additive manufacture is an additive manufacturing technique that stacks materials to manufacture a wide variety of products. It uses the properties of metal powder and the diversity of manufacture processes, to manufacture complex work pieces. This paper mainly discusses the selective laser melting (SLM) process, which uses high-power lasers melting metal powder in the processing area to reach the melting point. This leads to an instant melting and solidification with the multiple layers being melted and superimposed into a solid, to form a three-dimensional element. Nowadays, SLM produces products with excellent quality; the process of SLM is complex, however, with the quality of the finished product being affected by many external factors, including laser parameters, scanning speed, environmental condition, and material selection [1,2,3,4].

In this study, FeSiCr is proved to have lower core loss and eddy current loss at high frequency via the comparison analysis between FeSiCr and silicon steel sheets, a popularly used material for electric motor construction. However, the FeSiCr powder in the SLM process will appear in situations of uneven powder distribution, agglomeration and over-melting. Defect detection is required to avoid errors to improve the process yield.

Due to the above reasons, it is necessary to monitor the SLM process. The traditional flaw detection mainly integrates external devices, rather than computer vision. Craeghs uses a powder-spreading device to detect defects in the difference of speeds, between general metal powder and melting deformation [5]. Craeghs also uses the gray-scale value of the powder bed fusion (PBF) image to determine and evaluate the possible damage of recoater damaged [6]. Li uses different molten pool parameters to determine the current processing situation [7]. In other methods, Jacobsmuhlen uses X-rays for defect detection [8], while Kirka uses thermal signals as the basis for his defect detection’s method [9]. Cao uses light-emitting diode irradiation to obtain images with different incident angles for defect technique detection [10]. The objective of using optical irradiation is to obtain spectral images as well as obtaining the profile and porosity of the powder bed fusion [11,12].

Traditional detection methods usually lack efficiency and specificity. Thus, they need extra sensor devices such as X-ray or ultrasonic instruments in the requirements of detection of two or several types of defects. Lin uses traditional image processing for defect image segmentation, which are combined with multilayer perception (MLP) and a support vector machine (SVM) for defect detection [13]. This work is focused on several image classifications with increased impurity and scraper damage defects, reporting a classification accuracy of 98.33% and 97.5% for MLP [14] and SVM [15], respectively. Scime uses a multi-feature layer convolutional neural network for defect detection, resulting in the classification of large, medium, and small defects [16]. Later, the same authors improved this method in [17], introducing an additional Unet model [18] to the multi-feature classification proposed in [16], which improved results of defect classification by image segmentation. In [16,17], six different powder bed fusion anomaly classes are chosen: recoater hopping, recoater streaking, debris, super-elevation, part damage, and incomplete spreading. The resulting classification accuracy rate is 75% [16]; the test true positive rate is 0.84 [17]. These articles are limited to the image classifications with different types of defects without further discussions of defect location and segmentation.

In this paper, a two-stage deep learning model is proposed to detect three different types of defects. The purpose of the first stage is to classify images into different groups of with/without defects, by using the EfficientNet B7 model [19]. If one image belongs to the defective group, then it will be further segmented, and the defect’s type will be classified. The second stage uses Mask R-CNN [20], YOLO [21,22] and YOLACT [23] to the instance segmentation of the defect area pixel by pixel, and then selects the best one. The experimental results, as well as comparisons with other methods reveal that the used Mask R-CNN in the second stage is more precise and faster.

## 2. Materials and Methods

### 2.1. Experiment Setup

The images used in this study are recorded by the CCDs of the ITRI AM100 Laser-PBF equipment (Industrial Technology Research Institute (ITRI, Taiwan). The image recording device used in the AM100 Laser-PBF equipment is a GS3-PGE-91S6 (Teledyne FLIR LLC, Wilsonville, OR, USA) with an image size of 3376 × 2704 pixels^2^. The equivalent pixel 0.0374 mm/pixel can be obtained through camera calibration and perspective transformation. The computation time of image acquisition takes 0.078 s.

#### 2.1.1. Materials

The powder sample in this paper is manufactured by a commercial FeSiCr magnetic material. The powder’s chemical composition and particle-size distribution are listed in Table 1. The soft magnetic composite FeSiCr material is analyzed for its beneficial electromagnetic properties and unique 3D formation capacity [24]. A microscopic and spectroscopic experiment performed on FeSiCr after SLM process showed that FeO is the oxide layer responsible for the unique advantages of this material.

#### 2.1.2. Camera Calibration and Perspective Transformation

The aim of camera calibration is to establish the correspondence between world and image coordinates. More precisely, there is the need to determine the camera’s intrinsic matrix, extrinsic matrix, and distortion matrix. In general, extrinsic matrix converts world coordinates to camera coordinates and intrinsic matrix converts camera coordinates to image coordinates.

The association between image coordinates and world coordinates given by camera calibration is defined as Equation (1) [25].
(1)[xcyc1]==(αγx00βy0001)(R2∗2t2∗101)[xwyw1],
where (αγx00βy0001) and (R2∗2t2∗101) are the intrinsic and extrinsic matrices. The intrinsic matrix is composed of camera parameters, where xw, yw are world coordinate; xc, yc are camera coordinate; α and β are the scale factors in the image x and y axes, respectively, and γ is the parameter that describes the skewness of the two image axes. The extrinsic matrix is composed of the displacement and rotation required to convert the world coordinate into the camera coordinate, where R2∗2 is a 2 * 2 rotation matrix and t2∗1 is a displacement matrix.

Moreover, two kinds of distortions introduce significant distortion to images. One is radial distortion, which causes straight lines to curve, which is defined as Equation (2). The other is tangential distortion, which occurs because the image-taking lens is not aligned perfectly parallel to the imaging plane, defined as Equation (3).
(2){xrad_distorted=x(1+k1r2+k2r4+k5r6)yrad_distorted=x(1+k1r2+k2r4+k5r6),
(3){xtan_distorted=x+[2k3xy+k4(r2+2x2)]ytan_distorted=y+[k3(r2+2y2)+2k4xy],
where k1, k2, k3, k4, k5 are distortion parameter; these parameters are influenced by the extrinsic matrix and intrinsic matrix, and r2 = x2 + y2.

In this paper, we use a checkerboard as a calibrated board, which is shown in Figure 1b; the original image is shown in Figure 1a; and the calibrated image is shown in Figure 1c. Figure 1c shows one example after the image is calibrated, leading us to conclude that the image is still not flat. Therefore, we need convert it with a linear transformation matrix, which is shown as Equation (4).
(4)[x′y′] = M [xcyc],
where x′, y′ are the pair of corrected coordinates; *M* is a perspective transformation; and xc, yc denote for the pair of the original coordinate. The effect of perspective transformation is shown in Figure 1d.

#### 2.1.3. Defect Definition

Three major pieces of equipment affect SLM processing: the laser beam, the metal powder, and the recoater. Improper parameters of the laser beam will cause residual thermal stress that further results in warping, deformation, or breaking of the workpiece. These situations result in the occurrence of several defects in the process of powder-spreading. The defects are defined as the powder uncover class. When powder spreading happens, some areas are defined as the powder uneven class because of the uneven powder coverage. The large defective powder uncover causes damage in the re-coater as a result of generating the vertical scratches. The scratches are grouped into the recoater scratch class. To sum up, three different types of defects are defined as the powder uncover, as shown in Figure 2a; the powder uneven, as shown in Figure 2b; and the recoater scratch, as shown in Figure 2c.

### 2.2. Used Deep Learning Models

Two consecutive deep learning models are used to detect and locate several defects. The first stage is to determine whether the acquired images are with/without defects. One image without defect will be ignored (see Figure 2d); otherwise, it is further fed into the next stage to locate defects, so the three different types of defects can be located. The complete procedures are shown in Figure 3.

#### 2.2.1. Efficient Net

The EfficientNet [19] is used as the first model for classifying images with/without defects. In past deep learning models, only a one-dimensional parameter of depth, width or resolution is modified, which probably leads to the performance bottleneck in an ImageNet dataset. EfficientNet uses Equation (5) to find the optimal depth, width and resolution of neural networks by modifying multiple scales to achieve the best performance.
(5)N(d,w,r) = ⊙i=1…sFiLi(X〈Hi,Wi,Ci〉),
where FiLi denotes layer *F_i_* repeated *L_i_* times in stage *i*, and <*H_i_*, *W_i_*, *C_i_*> denotes the shape of input tensor *X* of the layer. The convolution net *N* is represented by a list of composed layers: *N* = Fk⊙…⊙F2⊙F1 and *s* denotes for the *s*-th layer. The computation method of Equation (5) aims to find the *H_i_*, *W_i_*, *C_i_* with maximum accuracy.

#### 2.2.2. Mask R-CNN

Mask R-CNN is a two-stage detection network that includes feature extraction model, region proposal network and prediction head shown in Figure 4. The feature extraction is implemented by using the feature pyramid network (FPN) [26] with backbones of ResNet 101 or ResNet 152 [27] in the first stage. In the second stage, the region proposal network (RPN) generates candidate objects according to the features; the region of interest (ROI) alignment finds the candidate objects proposed by RPN and the multi-scaled features of the backbone. Finally, the classification and regression branches of the prediction head infer the object position and classification, and the mask branch determines the mask of object and the corresponding bounding box.

#### 2.2.3. Loss Function for Mask RCNN

Several commonly used loss functions for classification, regression and mask in Mask R-CNN are defined in Equations (6)–(9).
(6)Lbox={0.5x2,|x|<1|x|−0.5,x<−1  or  x>1
(7)Lcls = −∑i=1nyilog(y^i)
(8)Lmask = −∑i=1ny^ilogyi+(1−y^i)log(1−yi)
(9)LMaskRCNN = Lbox + Lcls + Lmask
where Lbox and Lcls are same as faster R-CNN, Lbox is the loss bounding box regression, and Lcls is the loss for classification. Lmask is the binary cross entropy, which is used to classify whether the pixel is defective, yi is the target value, and y^i is the prediction value.

#### 2.2.4. Metrics

In this paper, mean average precision (*mAP*) and the Dice coefficient are employed to quantify the performance of algorithm. The precision, recall, average precision (*AP*) and *mAP* are defined as shown in Equations (10)–(13), respectively.
(10)precision = TPTP+FP
(11)recall = TPTP+FN
(12)AP = ∫01P(R)dR
(13)mAP = 1C∑i=1CAPi
where *TP* (true positive) denotes the number of defects that are correctly classified, *FP* (false positive) denotes the number of defects that are classified as normal, *FN* (false negative) denotes the normal objects that are identified as defects, and *C* is number of classes to be detected.

Next, we define the Dice coefficient as Equation (14), which can also be defined with a binary confusion matrix, that is *F*1 score, which is shown as Equation (15)
(14)Dice Coefficient = 2∗|pred|∩|gt||pred|+|gt|,
where *pred* denotes to the prediction mask and *gt* denotes to the ground truth.
(15)F1 = 2TP2TP+FP+FN

## 3. Experimental Results

In this section, three important system requirements are addressed as follows:The classification accuracy of images with/without defects must be more than 0.95.The computation time of the detection of three types of defects is limited to below 2.0 s.The performance of defect segmentation, Dice coefficient, between ground-truth and predicted defects, is at least more than 0.90.

### 3.1. Environment and Data

The training environment is described as follows:
Software: Linux Ubuntu 20.04 LTE, Python 3.9, Pytorch 3.10 with Detectron 2 [28];Hardware: Personal computer of Intel Core I9-10900K with main memory 64GB RAM and a Nvidia GeForce RTX 3090 GPU processor.

The whole datasets retrieved from ITRI are shown in Table 2. The proposed model mentioned in Section 3 has two stages: (1) the classification of the detecting image into with/without flaws classes, and (2) the segmentation of the corresponding defect areas.

### 3.2. Defected Image Classification

The used model is the Efficient Net B7; the test dataset is shown in Table 3. In this stage, we train about 50 epochs and use Adam as an optimizer. The learning rate is set to 0.001 initially, when training for 20 epochs, it drops to 0.0001. In total, the used number of parameters of Efficient Net B7 is 66,365,975. The results of the confusion matrix (see Table 4), and other performance metrics are listed in Table 5; the corresponding precision-recall (PR) curve is shown in Figure 5, which meet the system’s requirement.

### 3.3. Defects Segmentation

This study uses four-fold cross-validation method in datasets (see Table 6) to measure the defect detection and segmentation. We train about 60,000 iterations and define the anchors with IOU scores greater than 0.43 as positive. In addition, SGD method is used as an optimizer with learning rate 0.001 initially, which decay one tenth per 10,000 iterations. The size of anchor is set to [4, 8, 16, 48, 96, 216, 480, 720, 860]; the aspect ratio is defined as [0.1, 0.2, 0.5, 1, 2, 5, 10, 25, 50, 60, 70]. Four different CNN models: Mask RCNN with ResNet 101 backbone, Mask RCNN with Resent 152 backbone, YOLACT, and YOLOv3+Unet are used to segment the defects. The first three models are belonging to instance segmentation model, and the final one, YOLOv3+Unet model, uses the YOLOv3 to detect the defect areas, and the segment the defects using the Unet model. Table 7, Table 8, Table 9 and Table 10 show that performance indices of defect detection and segmentation. In these Tables, the Dice coefficient is used to measure the defect segmentation; the mAP, AP_uneven_, AP_uncover_, AP_scatch_ are means of precision of all defects, powder uneven, powder uncover, and re-coater scratch. The Dice coefficients and mAPs of the four methods are 0.8934 and 0.8627 (Mask RCNN with ResNet 101 backbone), 0.9438 and 0.9272 (Mask RCNN with ResNet 152 backbone), 0.8475 and 0.8544 (YOLACT), and 0.9342 and 0.9187 (YOLOv3)). The model of Mask-RCNN with ResNet 152 backbone is superior to other three methods. Figure 6 shows the experimental results of samples by using the Mask R-CNN predictions. 

## 4. Discussion and Conclusions

As shown in Table 5 and Table 8, the accuracy of and Dice coefficient of Mask RCNN with Resent 151 backbone meet system’s requirements. Moreover, the FPS of 8.61 s. is much faster than system’s expectations. In general, the average computation time of one image including the image classification and defect segmentation takes approximately 0.130 s. which is 7.80 FPS. Additionally, the time of image acquisition and data transition is about 0.0897 s. In total, the average computation time of one image is about 0.2197 s.

Additionally, Table 7, Table 8, Table 9 and Table 10 show that AP_Uncover_ does not perform well compared with other two different types of defects. The AP Uncover, resulting from improper residual thermal stress, exhibits small defects (lower than 32 pixel^2^). Although we use small anchors with sizes of 4 or 8 to overcome the small defect problems, the AP_Uncover_ only achieves 0.9046.

Compared with [13], our method can be directly applied without adjusting the brightness threshold and without using the morphology of dilation and erosion. Compared with [16,17], our method has better performance and efficiency, regardless of the yield of the pre-processing procedures.

In the future, the proposed method can be extended to implement a corrective procedure once the defects have been detected during the MAM process. Although the Mask RCNN with Resent 151 backbone model met the requirements of the system’s computational time and detection accuracy, the utilizations of other CNN models such as the EfficientDet [29] and CenterNet [30] to develop a more powerful model for the powder spreading process is still an interesting further research avenue.

## Figures and Tables

**Figure 1 materials-15-05662-f001:**
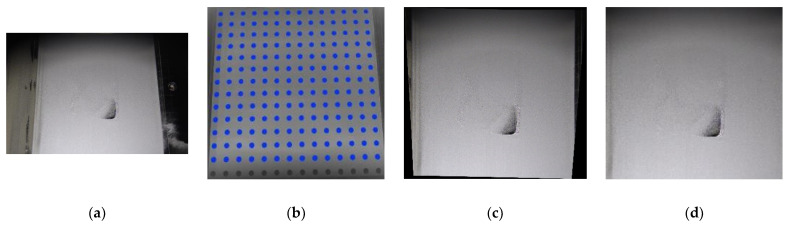
Camera calibration, (**a**) original image, (**b**) checkerboard for calibration, (**c**) image fetch from calibrated camera, (**d**) image after perspective transformation.

**Figure 2 materials-15-05662-f002:**
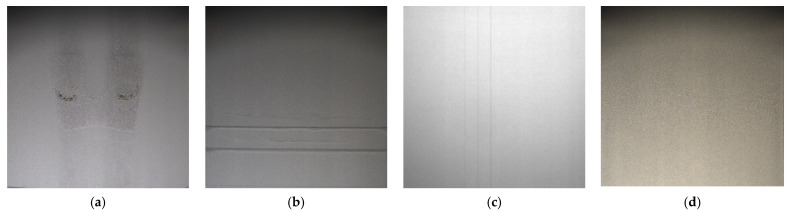
Three kinds of defects: (**a**) powder uncovered, (**b**) powder uneven, (**c**) recoater scratch and (**d**) flawless image. The flawless image is a normal case in the powder-spreading process.

**Figure 3 materials-15-05662-f003:**
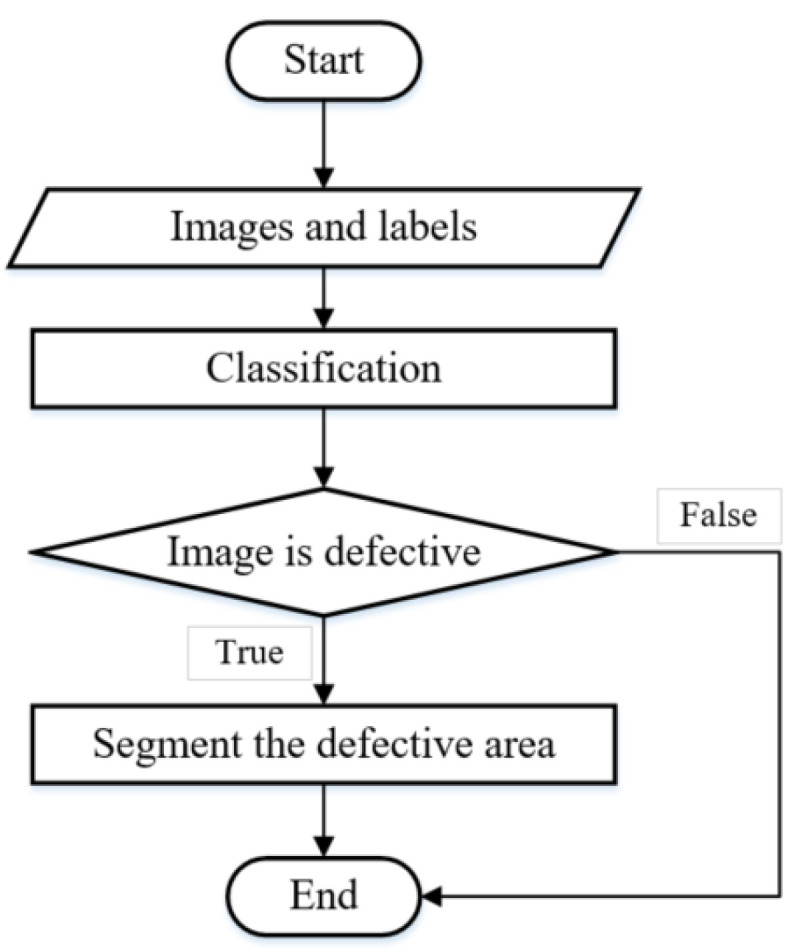
Flow Chart of Defects Detection System.

**Figure 4 materials-15-05662-f004:**
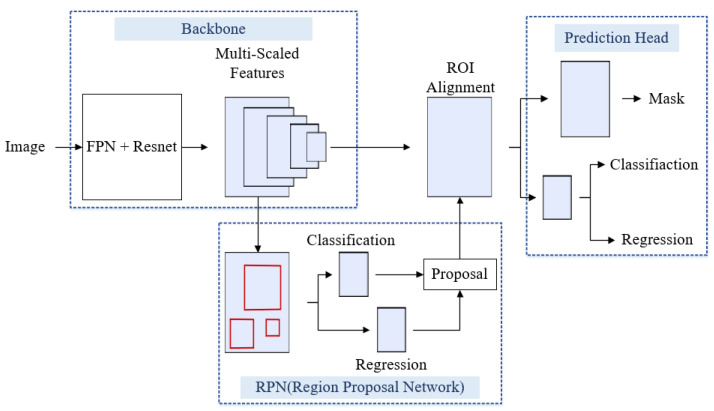
Mask R-CNN architecture.

**Figure 5 materials-15-05662-f005:**
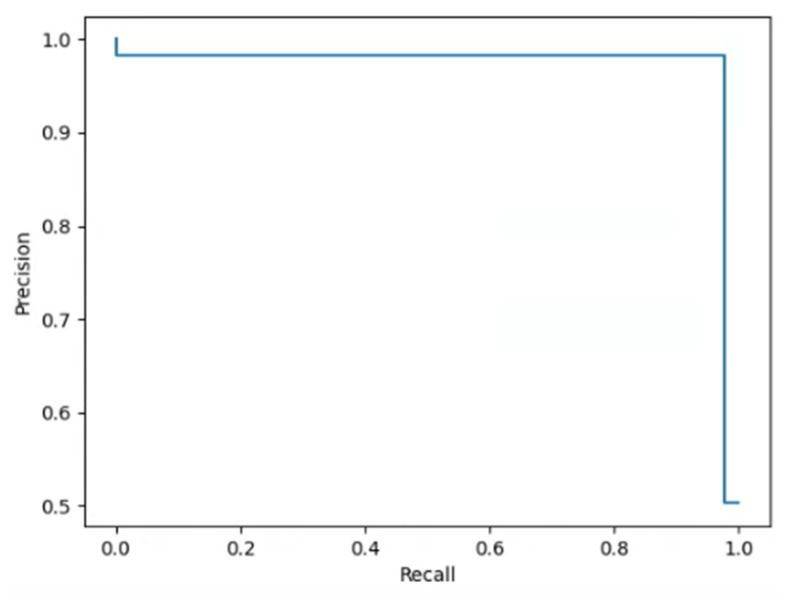
PR curve for image classification.

**Figure 6 materials-15-05662-f006:**
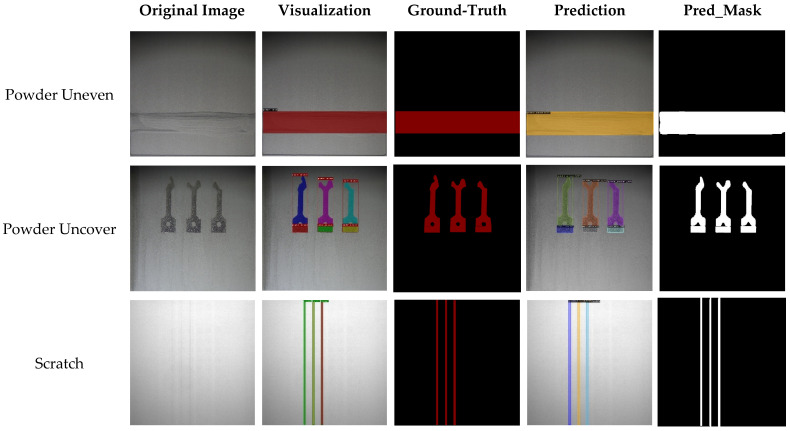
Samples for Mask R-CNN predictions.

**Table 1 materials-15-05662-t001:** Chemical composition and particle-size distribution of FeSiCr powder [21].

Materials	Chemical Composition		Particle Size Distribution (%)	
Fe-Si_3.5_-Cr_4.5_	Si (wt%)	3.47	D10 (μm)	15
	Cr (wt%)	4.44	D50 (μm)	34
	O (ppm)	2440	D90 (μm)	84

**Table 2 materials-15-05662-t002:** Statistics of all data.

Type	Total
Flawless (normal)	900
Powder Uncover	426
Powder Uneven	229
Recoater Scratch	239

**Table 3 materials-15-05662-t003:** Dataset Set Distribution for Classification.

Type	Total	Training Set	Validation Set	Test Set
Flawless (normal)	900	540	180	180
Defect	894	538	178	178

**Table 4 materials-15-05662-t004:** Confusion Matrix for Image Classification.

Confusion Matrix		Ground Truth	
		Normal	Defects
**Prediction**	**Normal**	178	1
	**Defects**	2	177

**Table 5 materials-15-05662-t005:** Performance Metrics of the Proposed Methodology.

Metrics	Accuracy	TP	Precision	Recall	F1 Score	FPS
Value	99.16	98.89	99.45	98.91	99.16	71.91

**Table 6 materials-15-05662-t006:** Cross Validation Dataset Set Distribution for Segmentation.

Fold	Fold 1	Fold 2	Fold 3	Fold 4
Images	245	245	245	244

**Table 7 materials-15-05662-t007:** Four-fold cross validation result, where FPS denotes for frame per second by using Mask-RCNN with ResNet 101 backbone. In total, the number of parameters Mask RCNN used is 69,188,563.

Fold	Dice (%)	mAP (%)	AP50(%)	AP75(%)	AP_uneven_(%)	AP_uncover_(%)	AP_scratch_(%)	FPS
Fold 1	91.24	88.47	96.65	94.81	92.17	79.94	97.87	9.27
Fold 2	88.32	84.48	96.83	93.63	91.67	78.47	98.47	9.27
Fold 3	90.42	87.64	97.68	96051	90.47	77.46	96.52	9.27
Fold 4	87.38	84.49	96.48	92.49	92.05	78.85	98.14	9.27
Average	89.34	86.27	96.91	94.36	91.59	78.68	97.75	9.27

**Table 8 materials-15-05662-t008:** Four-fold cross validation result, where FPS denotes for frame per second by using Mask-RCNN with ResNet 152 backbone. In total, the number of parameters used is 101,188,563.

Fold	Dice (%)	mAP (%)	AP50(%)	AP75(%)	AP_uneven_(%)	AP_uncover_(%)	AP_scratch_(%)	FPS
Fold 1	95.78	93.84	98.20	97.34	93.79	91.25	98.91	8.61
Fold 2	93.13	92.89	98.75	95.72	92.46	89.98	97.92	8.61
Fold 3	93.98	92.37	99.67	98.26	94.12	90.98	98.17	8.61
Fold 4	94.93	91.78	89.60	96.60	94.51	89.63	96.92	8.61
Average	94.38	92.72	98.97	96.98	93.72	90.46	97.98	8.61

**Table 9 materials-15-05662-t009:** Four-fold cross validation result, where FPS denotes for frame per second by using the YOLACT model. In total, the number of parameters used is 43,286,432.

Fold	Dice (%)	mAP (%)	AP50(%)	AP75(%)	AP_uneven_	AP_uncover_	AP_scratch_	FPS
Fold 1	86.18	87.50	94.20	94.78	85.71	83.76	92.87	19.94
Fold 2	83.22	83.90	92.71	88.42	82.36	80.35	89.57	19.94
Fold 3	87.16	85.37	94.97	93.56	87.62	81.72	91.77	19.94
Fold 4	82.44	80.99	93.93	93.88	86.15	80.01	88.87	19.94
Average	84.75	84.44	93.81	92.67	85.46	81.46	90.43	19.94

**Table 10 materials-15-05662-t010:** Four-fold cross validation result, where FPS denotes for frame per second by using the YOLOv3+Unet model. In total, the number of parameters used is 69,537,100. The YOLOv3 detects the defect area, and then the Unet segment the defects.

Fold	Dice (%)	mAP (%)	AP50(%)	AP75(%)	AP_uneven_	AP_uncover_	AP_scratch_	FPS
Fold 1	94.21	92.84	99.01	94.79	92.69	89.67	97.87	16.6
Fold 2	93.67	91.67	98.98	94.01	91.67	89.99	98.00	16.6
Fold 3	93.19	90.62	97.94	95.67	90.78	90.47	96.14	16.6
Fold 4	92.61	92.35	98.79	92.93	91.91	89.35	98.35	16.6
Average	93.42	91.87	98.68	94.35	91.76	89.87	97.59	16.6

## Data Availability

The data that support the finding of this study are available from the corresponding author upon reasonable request.

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
