# Peer review of "Deep Learning Applied to Defect Detection in Powder Spreading Process of Magnetic Material Additive Manufacturing"

_materials, 2022, doi:10.3390/ma15165662_

Round 1
Reviewer 1 Report
In the manuscript, the authors designed a deep learning system to detect possible defects in the SLM process. FeSiCr powder was adapted to prepare the data set can be detected by cameras. On the whole, it's an effective attempt to apply deep learning into the powder spreading process, and it may help in industry.
However, I still have some concerns about the deep learning part:
1. Samples in Figure 2 have distinct differences on background. This is a significant factor will affect the performance of the deep learning models. The authors should explain the way they process backgrounds in detail.
2. There is no direct comparison with the other approaches, although the authors mentioned them briefly in Introduction and Discussion parts. The lack of quantitative comparisons will weaken the feasibility of the method authors proposed. Thus, a baseline model is necessary, no matter it is from the expert experiences or existing models, e.g., U-Net.
3. EfficientDet is also a suitable model for the segmentation task, and it's a natural generalization of the EffcientNet. The authors should at least discuss it in the final part.
4. The comparison in figure 6 seems trivial, because patterns in Power Uneven and Scratch can even be recognized by naked eyes easily. Even if the Powder Uncover case is non-trivial, the model didn't achieve a satisfying result. The authors should at least explain details in the caption of Figure 6.
The above concerns should be replied appropriately before possible acceptance.
Author Response
Reviewer 1
However, I still have some concerns about the deep learning part:
Comment 1. Samples in Figure 2 have distinct differences on background. This is a significant factor will affect the performance of the deep learning models. The authors should explain the way they process backgrounds in detail.
Reply: The authors are appreciative of the reviewer’s comments. The description of how the backgrounds were processed is described in Section 2.2 of the revised manuscript (lines 148-153, page 4). Only flaw images are used to detect and segment the defects.
Comment 2. There is no direct comparison with the other approaches, although the authors mentioned them briefly in Introduction and Discussion parts. The lack of quantitative comparisons will weaken the feasibility of the method authors proposed. Thus, a baseline model is necessary, no matter it is from the expert experiences or existing models, e.g., U-Net.
Reply: The authors are thankful of the reviewer’s comments. The Four different CNN models, Mask RCNN with ResNet 101 backbone, Mask RCNN with ResNet152 backbone, YOLACT, and YoLov3 were compared (see Tables 7-10). Details of the comparison are listed in Section 3.3 (lines 231-248, page 8)
Comment 3. EfficientDet is also a suitable model for the segmentation task, and it's a natural generalization of the EffcientNet. The authors should at least discuss it in the final part.
Reply: Thank you for the suggestion. The EfficientDet is included in the Discussion and Conclusion Section of the revised manuscript. (see lines 288 - 293, page 10)
Comment 4. The comparison in figure 6 seems trivial, because patterns in Power Uneven and Scratch can even be recognized by naked eyes easily. Even if the Powder Uncover case is non-trivial, the model didn't achieve a satisfying result. The authors should at least explain details in the caption of Figure 6.
Reply: The authors appreciate the reviewer’s suggestion. The explanation about the difficulty in detecting the Powder Uncover defects is included in the Discussion and Conclusion Section of the revised manuscript (lines 272-278, page 10).

Reviewer 2 Report
The authors propose a DL based Defect Detection algorithm in Powder Spreading Process of Magnetic Material Additive Manufacturing. The current study is interesting. In general, the main conclusions presented in the paper are supported by the figures and supporting text. However, to meet the journal quality standards, the following comments need to be addressed.
1. Abstract: Should be improved and extended. The authors talk lot about the problem formulation, but novelty of the proposed model is missing. Also provided the general applicability of their model. Please be specific what are the main quantitative results to attract general audiences.
2. The introduction can be improved. The authors should focus on extending the novelty of the current study. Emphasize should be given in improvement of the model (in quantitative sense) compared to existing state-of-the art models.
3. More details about network architecture and complexity of the model should be provided.
4. what about comparison of the result with current state-of-the art models? Did authors perform ablation study to compare with different models?
5. What are the baseline models and benchmark results? The authors can compared the result with existing models evaluated with datasets
6. Conclusion parts needs to be strengthened.
7. Please provide a fair weakness and limitation of the model, and how it can be improved.
8. Typographical errors: There are several minor grammatical errors and incorrect sentence structures. Please run this through a spell checker.
9. Following references can be added as relevant deep learning object detection references ( see : Scientific Reports 11, 1447 (2021) https://doi.org/10.1038/s41598-021-81216-5’ Neural, Comput & Applic (2022) https://doi.org/10.1007/s00521-021-06651-x. Hence they can be discussed in the related work section.
Author Response
Reviewer 2.
Comment 1. Abstract: Should be improved and extended. The authors talk lot about the problem formulation, but novelty of the proposed model is missing. Also provided the general applicability of their model. Please be specific what are the main quantitative results to attract general audiences.
Reply: The authors are grateful for the reviewer’s comments. Base on the suggestions made by the reviewer, the abstract of the revised manuscript has been significantly improved and extended to include the novelty and main contribution of the paper (lines 22-27, page 1).
Comment 2. The introduction can be improved. The authors should focus on extending the novelty of the current study. Emphasize should be given in improvement of the model (in quantitative sense) compared to existing state-of-the art models.
Reply: The authors appreciate the reviewer’s suggestions. The introduction section has been improved with emphasis on the comparison between the current study and existing state-of-the-art models (lines 76-83 of page 2). The comparisons of the four models studied are listed in Tables 7-10.
Comment 3. More details about network architecture and complexity of the model should be provided.
Reply: Thank you for the comments. The details of the best model of Mask RCNN have been described in the Section 2.2.2 (lines 170-178) of the revised manuscript. The used parameters of the four models are shown in Tables 7-10.
Comment 4. what about comparison of the result with current state-of-the art models? Did authors perform ablation study to compare with different models?
Reply: The authors are grateful for the insightful questions. (see Tables 7-10) This study compares the four existing CNN models and then selected the best model for defect detection in the powder spreading process. The four different CNN models that were compared are Mask RCNN with ResNet 101 backbone, Mask RCNN with ResNet152 backbone, YOLACT, and YOLOv3. Details of the comparison are listed in Section 3.3 (lines 230-248, page 8).
Comment 5. What are the baseline models and benchmark results? The authors can compared the result with existing models evaluated with datasets
Reply: The authors appreciate the reviewer’s thoughtful comments. The following four different CNN models were compared: Mask RCNN with ResNet 101 backbone, Mask RCNN with ResNet152 backbone, YOLACT, and YoLov3. The details of the comparison are listed in Section 3.3 (see Tables 7-10).
Comment 6. Conclusion parts needs to be strengthened.
Reply: The authors appreciate the reviewer’s comment. The conclusion has been improved to significantly provide a logic summary (lines 272-293, page 10).
Comment 7. Please provide a fair weakness and limitation of the model, and how it can be improved.
Reply: The authors are grateful for the reviewer’s suggestion. The weakness and limitation of the Mask RCNN in the defect detection were stated in the Discussion and Conclusion Section. The possible improved approaches are described in lines 288-293.
Comment 8. Typographical errors: There are several minor grammatical errors and incorrect sentence structures. Please run this through a spell checker.
Reply: The authors are grateful for the reviewer’s comment. The revised manuscript has been carefully checked by the an English editing firm, ServiceScape, url:(https://www.servicescape.com/).
Comment 9. Following references cabe added as relevant deep learning object detection references ( see : Scientific Reports 11, 1447 (2021) https://doi.org/10.1038/
s41598-021-81216-5’ Neural, Comput & Applic (2022) https://doi.org/10.1007/
s00521-021-06651-x. Hence they can be discussed in the related work section.
Reply: The authors appreciate the reviewer’s suggestion. The suggested papers have been discussed in the Introduction section (lines 80-83) and subsequently added in the reference list (Ref. 21 and 22).

Reviewer 3 Report
I think that the paper should have covered more the issues of materials and the process of laser printing. The paper seems to me to be more for another magazine (which deals with graphics), although I notice that the cited paper 13 has already been published in Materials, where similar content is covered.
In general, my opinion is that the paper is outside the scope of the journal Materials. The essential problem of the work is image detection. The application of this detection is in the field of SLM (Selective Laser Melting). The paper is correct, but in my opinion for a magazine that deals with graphics.
The paper deals with image detection, based on which it determines whether there is a defect in the material during the SLM process. If it exists, it is divided into three types of defects. I'm not that good at image detection, so I can't judge. It can certainly be applied to solids, including different materials. Image detection was not applied when determining defects in the SLM process. In my opinion, it is paper for a magazine from graphic processing and I believe it will be published. I have no objections to the scientific contribution. Scientific contribution is evident in the field of image detection.
Author Response
Reviewer 3.
Comment. The paper deals with image detection, based on which it determines whether there is a defect in the material during the SLM process. If it exists, it is divided into three types of defects. I'm not that good at image detection, so I can't judge. It can certainly be applied to solids, including different materials. Image detection was not applied when determining defects in the SLM process. In my opinion, it is paper for a magazine from graphic processing and I believe it will be published. I have no objections to the scientific contribution. Scientific contribution is evident in the field of image detection.
Reply: The authors appreciate the reviewer’s opinion. The powder spreading stage of the selective laser melting process always generates several defects such as powder uncover, powder uneven and scratch, but the accumulation of defects leads to failures of products. Early defect detection and recovery are very important to avoid failures during magnetic material additive manufacturing. However, defect detection is a challenging job. Recently, the deep learning model has been widely applied to detect interesting medical problems such as lesion detection; therefore, this paper tries to detect detection in the powder spreading process by using the CNN model. Although the technology presented in this paper focuses on the usage of the deep learning model, this application is still in the SLM process of magnetic material additive manufacturing. As a result, the authors believe it’s appropriate to publish this study in the Materials Journal if accepted.

Reviewer 4 Report
The authors proposed the following manuscript: "Deep Learning Applied to Defect Detection in Powder Spreading Process of Magnetic Material Additive Manufacturing " a manuscript that is generally well written, well structured and documented.
The introduction provides sufficient information about the study, the experimental part and the results are well constructed. However, the Discussion part should be enriched or the Conclusions chapter could be introduced where the authors could discuss the results of their work in more detail.
Author Response
Reviewer 4.
Comment. The introduction provides sufficient information about the study, the experimental part and the results are well constructed. However, the Discussion part should be enriched or the Conclusions chapter could be introduced where the authors could discuss the results of their work in more detail.
Reply: The authors are grateful for the reviewer’s comments. The Discussion and Conclusion Section of the revised manuscript has been significantly improved to include a summarized detail of the results presented in this study.

Round 2
Reviewer 1 Report
The authors answer all my questions or concerns properly, thus, I recommend the publication.
Reviewer 2 Report
The revised manuscript is suitable for publication in Materials.